# Immunological and Microbiological Profiling of Cumulative Risk Score for Periodontitis

**DOI:** 10.3390/diagnostics10080560

**Published:** 2020-08-05

**Authors:** Joonas Liukkonen, Ulvi K. Gürsoy, Eija Könönen, Ramin Akhi, Aino Salminen, John M. Liljestrand, Pratikshya Pradhan-Palikhe, Milla Pietiäinen, Timo Sorsa, G. Rutger Persson, Päivi Mäntylä, Kåre Buhlin, Susanna Paju, Juha Sinisalo, Sohvi Hörkkö, Pirkko J. Pussinen

**Affiliations:** 1Institute of Dentistry, University of Turku, 20100 Turku, Finland; ulvi.gursoy@utu.fi (U.K.G.); eija.kononen@utu.fi (E.K.); 2Oral Health Care, Welfare Division, City of Turku, 20100 Turku, Finland; 3Medical Microbiology and Immunology, Research Unit of Biomedicine, University of Oulu, 90100 Oulu, Finland; ramin.akhi@oulu.fi (R.A.); sohvi.horkko@oulu.fi (S.H.); 4Medical Research Center, Oulu University Hospital and University of Oulu, 90100 Oulu, Finland; 5Nordlab, Oulu University Hospital, 90100 Oulu, Finland; 6Oral and Maxillofacial diseases, University of Helsinki and Helsinki University Hospital, 00100 Helsinki, Finland; aino.m.salminen@helsinki.fi (A.S.); john.liljestrand@helsinki.fi (J.M.L.); pratikshya.pp@gmail.com (P.P.-P.); milla.pietiainen@helsinki.fi (M.P.); timo.sorsa@helsinki.fi (T.S.); paivi.mantyla@uef.fi (P.M.); kare.buhlin@ki.se (K.B.); susanna.paju@helsinki.fi (S.P.); 7Department of Periodontology, Institute of Odontology, Karolinska Institutet, 12343 Huddinge, Sweden; 8Departments of Periodontics and Oral Medicine, University of Washington, Seattle, WA 98101, USA; rper@uw.edu; 9Department of Periodontology, University of Bern, 3001 Bern, Switzerland; 10Institute of Dentistry, University of Eastern Finland, 70100 Kuopio, Finland; 11Oral and Maxillofacial Diseases, Kuopio University Hospital, 70100 Kuopio, Finland; 12HUCH Heart and Lung Center, Helsinki University Central Hospital, 00100 Helsinki, Finland; juha.sinisalo@hus.fi

**Keywords:** biomarkers, bacteria, lipopolysaccharides, saliva, serum

## Abstract

The cumulative risk score (CRS) is a mathematical salivary diagnostic model to define an individual’s risk of having periodontitis. In order to further validate this salivary biomarker, we investigated how periodontal bacteria, lipopolysaccharide (LPS), and systemic and local host immune responses relate to CRS. Subgingival plaque, saliva, and serum samples collected from 445 individuals were used in the analyses. Plaque levels of 28 microbial species, especially those of *Aggregatibacter actinomycetemcomitans*, *Porphyromonas gingivalis*, *Porphyromonas endodontalis*, *Prevotella intermedia*, and *Tannerella forsythia*, and serum and salivary levels of IgA and IgG against these five species were determined. Additionally, LPS activity was measured. High CRS associated strongly with all IgA/IgG antibody and LPS levels in saliva, whereas in serum the associations were not that obvious. In the final logistic regression model, the best predictors of high CRS were saliva IgA burden against the five species (OR 7.04, 95% CI 2.25–22.0), IgG burden (3.79, 1.78–8.08), LPS (2.19, 1.38–3.47), and the sum of 17 subgingival Gram-negative species (6.19, 2.10–18.3). CRS is strongly associated with microbial biomarker species of periodontitis and salivary humoral immune responses against them.

## 1. Introduction

Periodontitis, an infection-driven chronic inflammation in tooth-supporting tissues, is highly common in adults and a major cause of tooth loss. High levels of periodontitis-associated subgingival bacteria and the humoral immune response to periodontal infection are considered important factors for the initiation and progression of periodontal disease [1]. In addition to dysbiosis-associated Gram-negative periodontal pathogens *Aggregatibacter actinomycetemcomitans*, *Porphyromonas gingivalis*, *Prevotella intermedia*, and *Tannerella forsythia* [2,3], increasing knowledge of polymicrobial synergy and the development of sequencing techniques have resulted in the discovery of new potential periodontal pathogens, e.g., *Porphyromonas endodontalis* [4,5]. Lipopolysaccharide (LPS) of Gram-negative bacteria is a classical trigger for septic shock, and an independent risk factor for chronic cardiometabolic disorders [6]. Circulating LPS leads to systemic low-grade inflammation [7]. It has been suggested that a significant part of the circulating LPS is originated from the oral cavity and that circulating LPS activity may be a link between periodontitis and coronary artery disease (CAD) [8].

Besides detecting periodontal pathogens directly in saliva or subgingival plaque, the odds for the pathogen’s presence can be evaluated by measuring the levels of serum or saliva antibodies [1,9,10,11]. Immunoglobulin A and G (IgA/IgG) are usually determined in this context. Elevated levels of systemic antibodies against periodontal bacteria have been found in individuals with periodontitis [11,12,13,14,15,16] and, thus, serum antibody levels are considered potential diagnostic markers of the disease. According to our previous studies, IgA and IgG levels in serum associate with *P. gingivalis* and *A. actinomycetemcomitans* carriage in saliva, and the combination of serum antibody levels and salivary pathogen levels improves the detection of periodontitis [11,17,18]. Due to the local production of antibodies, the antibody levels against periodontal bacteria are higher in gingival crevicular fluid than in serum. Elevated IgA levels against *A. actinomycetemcomitans* and *P. gingivalis* [10] and IgG levels against *A. actinomycetemcomitans* leukotoxins [19] have been observed in the saliva of individuals suffering from periodontitis.

The fluctuating nature of periodontal pathogenesis limits the use of single biomarkers with established thresholds in the detection of periodontitis [20]. The cumulative risk score (CRS) is a mathematical model to define an individual’s risk of having periodontitis [21]. Three selected salivary biomarkers, namely, *P. gingivalis*, interleukin (IL)-1β, and matrix metalloproteinase (MMP)-8, represent three components of the periodontal inflammatory process: microbial pathogen burden, cytokine production, and tissue degradation, respectively. Our previous results indicate that high CRS is significantly associated with clinical and radiological signs of periodontitis [21,22,23]. In the present study, our aim was to validate further the CRS by analyzing its associations with subgingival microbial biomarker species and salivary humoral immune responses against them.

## 2. Materials and Methods

### 2.1. Study Population

The Corogene study is a prospective cohort including Finnish patients who underwent coronary angiography between June 2006 and March 2008 at the Helsinki University Central Hospital [24]. The Parogene is a sub-study of 508 subjects, who received clinical and radiographic oral health examinations. As described in detail elsewhere [25,26], these examinations were performed by calibrated dental specialists. The information of dental hygiene and smoking habits were gathered with a questionnaire. All subjects signed an informed consent, and the study was approved by the Helsinki University Central Hospital ethics committee (approval reference number 106/2007, approved on the 1 August 2007). The present study population consists of the Parogene study patients based on the availability of their saliva determinations (*n* = 445, 87.6% of the Parogene cohort).

### 2.2. Serum and Saliva Samples

Serum samples were drawn from the arterial line during the coronary angiography and stored at −80 °C until further analyses [24]. Saliva samples were collected before the oral examination between 8 a.m. and 3 p.m. Study participants chewed a piece of paraffin for 5 min, and at least 2 mL of stimulated whole saliva was collected and stored at −70 °C until further use. The samples were analyzed blindly in the laboratory. After thawing, the samples were centrifuged at 9300× *g* for 5 min. The pellets from the process were used for the bacterial analyses, whereas the supernatant was used for the antibody determinations.

### 2.3. Measurement of Serum and Saliva Antibodies Against Periodontal Pathogens

An enzyme-linked immunosorbent assay (ELISA) [11,27] was used to determine serum levels of IgA and IgG against whole cell antigens of *A. actinomycetemcomitans*, *P. gingivalis*, *P. endodontalis*, *P. intermedia*, and *T. forsythia.* The details of these analyses are described elsewhere [18,28]. Salivary levels of IgA/IgG against the same whole cell antigens of the same species were analyzed using a chemiluminescence immunoassay as described in detail elsewhere [29]. The antigens used in the assays, dilutions, and inter-assay variations are presented in Appendix A.

### 2.4. Subgingival Bacteria

Pooled subgingival plaque samples were collected from the deepest pocket of each jaw quadrant [30]. Checkerboard DNA–DNA hybridization was used to determine the levels of 28 bacterial species, and of these, the levels of the same five species used for the antibody determinations were selected for detailed analyses [31,32]. Additionally, a sum level of 17 Gram-negative (*P. gingivalis*, *T. forsythia*, *T. denticola*, *A. actinomycetemcomitans* (strains Y4 plus ATCC 29523), *P. intermedia*, *Fusobacterium nucleatum* ssp. *naviforme*, *F. nucleatum* ssp. *nucleatum*, *F. nucleatum* ssp. *polymorphum*, *F. periodonticum*, *Campylobacter rectus*, *C. gracilis*, *C. showae*, *Veillonella parvula*, *Capnocytophaga gingivalis*, *C. sputigena*, *C. ochracea*, and *Eikenella corrodens*) and 11 Gram-positive species (*Parvimonas micra*, *Actinomyces odontolyticus*, *A. israelii*, *A. naeslundii*, *A. neuii*, *Streptococcus mitis*, *S. oralis*, *S. sanguinis*, *S. gordonii*, *S. intermedius*, and *S. constellatus*) [32] were used in the statistical analyses.

### 2.5. LPS Activity in Serum and Saliva Samples

After diluting the samples with endotoxin-free water, the LPS activity was determined from the serum samples (1:5, *v*/*v*) and the saliva supernatant (1:40.000, *v*/*v*) by a commercially available assay, Limulus amebocyte lysate coupled with a chromogenic substrate (Hycult Biotec, Uden, The Netherlands). The inter-assay coefficient of variation was 5.5% for serum and 6.8% for the saliva samples.

### 2.6. CRS Determination

The CRS values of the present study population have been described previously [23]. Briefly, the concentrations of IL-1β and aMMP-8 were determined from the saliva supernatants and the quantity of *P. gingivalis* from the saliva pellets. IL-1β concentrations were determined with a flow cytometry-based technique (MILLIPLEX^®^ Map Kit, Human Cytokine/Chemokine panel MPXHCYTO-60K, Millipore, Billerica, MA, USA and Luminex^®^ xMAP™ technique, Luminex Corporation, Austin, TX, USA), and MMP-8 concentrations with a time-resolved immunofluorometric assay (IFMA), as described by Gürsoy et al. [33] A qPCR-based technique was used to quantitate *P. gingivalis* in saliva and the concentrations were calculated as genomic equivalents (GE)/mL [28]. All concentrations were divided into tertiles to achieve a cumulative sub-score for each study subject. These sub-scores were further multiplied with each other and used to categorize the study population with “low (CRS I), moderate (CRS II), or high (CRS III) risk of having periodontitis”. A detailed methodological description of CRS and its diagnostic abilities to detect periodontitis has been published elsewhere [21,23].

### 2.7. Statistical Analyses

Statistical analyses were performed with the statistical program (SPSS version 21.0; SPSS Inc., Chicago, IL, USA). The data were not normally distributed and therefore are presented in medians with interquartile range (IQR). The chi-square test was used when examining the relation of gender, proportion of edentulous, ever smoked, and diabetes mellitus (type 1 and 2) to CRS I–III. The Jonckheere–Terpstra test was used when examining the linear relation of CRS I–III with age, number of teeth, serum or saliva IgA/IgG, LPS activity, and subgingival bacterial levels. Logistic regression models were used when the association of high CRS (III vs. CRS I–II) with serum or saliva IgA/IgG and LPS activity, and subgingival bacteria levels was examined. Firstly, the models were adjusted for age, gender, and number of remaining teeth, and, if found significant, they were additionally adjusted for the subgingival level of the corresponding bacterial species. The associations of high CRS (CRS III vs. CRS I–II) with antibody, LPS, and bacterial levels were analyzed in multiple logistic regression models adjusted for age, sex, number of teeth, and smoking (ever vs. never). To standardize the values, all models were performed for logarithmically (10-base) transformed results. The *p*-value < 0.05 was considered significant.

## 3. Results

Table 1 shows the characteristics of the population in relation to CRS I–III. As expected, there were statistically significant differences in the number of teeth and periodontal status between the CRS groups.

Serum antibody and LPS activity levels in relation to CRS I–III are presented in Table 2. Elevating IgA levels against *P. gingivalis*, *P. endodontalis*, and *P. intermedia* and elevating IgG levels against *P. gingivalis*, *P. endodontalis*, and *P. intermedia* were found with increasing CRS, whereas the *T. forsythia* antibody levels or serum LPS activity did not differ between CRS I–III. In regression analyses, after adjusting for age, sex, and number of teeth (Model 1), high CRS (III) associated significantly with the IgA levels against *A. actinomycetemcomitans*, *P. gingivalis*, *P. endodontalis*, and *P. intermedia* and the IgG levels against *P. gingivalis*, *P. endodontalis*, and *P. intermedia*. When the models were further adjusted for the corresponding subgingival bacterial level, *A. actinomycetemcomitans* IgA, *P. gingivalis* IgG, *P. endodontalis* IgA/IgG, and *P. intermedia* IgA remained significantly associated with CRS III.

Saliva antibody and LPS activity levels in relation to CRS are presented in Table 3. The IgA/IgG antibody levels against all tested bacteria and LPS activity were elevating with increasing CRS. In logistic regression analyses adjusted for age, sex, and number of teeth (Model 1), high CRS (III) associated significantly with all IgA/IgG and LPS activity levels. These associations remained significant when the models were adjusted further for the corresponding subgingival bacterial level (the antibodies) or the sum of Gram-negative bacteria (LPS).

Subgingival bacterial levels in relation to CRS I–III are presented in Table 4. The counts of all examined bacteria, and the sum of 11 subgingival Gram-positive and 17 subgingival Gram-negative species increased with increasing CRS. In the association analyses adjusted for age, gender, and number of remaining teeth, high CRS (III) associated significantly with all examined bacterial species, and with the sum of 11 subgingival Gram-positive and 17 subgingival Gram-negative species.

Table 5 summarizes the associations of high CRS (III) with antibody and bacterial levels in a regression model that was adjusted for age, gender, number of teeth, and smoking. Salivary IgA burden against the five species, salivary IgG burden against the five species, salivary LPS, and the sum of subgingival Gram-negative species associated with high CRS. The odds ratios were 7.04 (*p* = 0.001), 3.79 (*p* = 0.001), 2.19 (*p* = 0.001), and 6.19 (*p* = 0.001), respectively.

## 4. Discussion

Here we present a strong association of salivary IgA burden, salivary IgG burden, salivary LPS, and the sum of subgingival Gram-negative species with high CRS. Due to its associations with subgingival periodontal bacteria and the salivary humoral immune response to them, the present findings produce additional evidence on the diagnostic strength of CRS.

Our study shows for the first time that LPS activity in saliva associates with a risk of having periodontitis, based on CRS. We used the Limulus Amebocyte lysate assay to measure LPS activity in saliva and serum, because it is the most sensitive and most widely used method [34]. The assay is not targeted to any specific bacterial species but determines the overall biological activity of a mixture of endotoxins produced by Gram-negative bacteria. It has been shown that serum LPS correlates with periodontal status and serum LPS activity decreases after periodontal treatment [35,36,37]. A significant part of circulating LPS may originate from the oral cavity. There is supporting evidence that there is a correlation between serum and saliva LPS activity levels [8]. Moreover, a study performed with mice estimated that oral administration of *P. gingivalis* could change the gut microbiome and lead to elevated serum endotoxin levels [38]. In investigations on structural alterations in lipid-A-derived 3-OH fatty acid profiles or using a bioassay for Toll-like receptors 2 and 4, stimulants indicate that there are differences in saliva between individuals with and without periodontal disease [39,40]. These methods differ from the LAL assay and do not measure the activity level of LPS. Based on our previous data on LPS activity in saliva, LPS activity correlates with the number of remaining teeth and the amount of alveolar bone loss; however, no relation to clinical periodontal parameters was found [8,28,41]. Similarly, in the present study, no association between serum LPS and CRS was observed.

To date, findings on IgA/IgG binding to periodontal bacteria in saliva are limited to *A. actinomycetemcomitans*, *P. gingivalis*, *T. denticola*, *P. intermedia*, and *F. nucleatum* [10,42,43,44]. In the present population, we also determined the salivary antibodies against *T. forsythia* and *P. endodontalis* [28,45]. Elevated salivary levels of IgA antibodies against *A. actinomycetemcomitans* and *P. gingivalis* [10] and salivary IgG levels against *A. actinomycetemcomitans* leukotoxins [19] have been detected in periodontitis patients. Since all determined salivary antibody levels associated with CRS, we calculated their sum to be used in further statistical analyses and to strengthen the association. Notably, salivary IgA/IgG burden against these microbial biomarker species presented an independent and strong association with CRS even in models where the bacterial subgingival levels were used as covariates. This indicates immune hyperresponsiveness to dysbiosis, typical for the pathogenesis of periodontitis [46].

Elevated levels of systemic antibodies against periodontal bacteria are common in periodontitis patients [12,14,15,16,47]. Conceivably, it can be assumed that there is a connection between CRS and systemic IgA/IgG response. In this study, high CRS (III) associated significantly with the serum IgA levels against *A. actinomycetemcomitans*, *P. endodontalis*, and *P. intermedia* and with the IgG levels against *P. gingivalis* and *P. endodontalis*. Based on the association analyses, these serum antibody levels were also independent of the subgingival bacterial level. Interestingly, the most significant association with high CRS was observed with antibodies against *P. gingivalis* and *P. endodontalis.* An obvious explanation is that CRS already includes *P. gingivalis* concentration and, in the case of *P. endodontalis*, cross-reactivity of antibodies to *P. endodontalis* and closely related species cannot be excluded. The results regarding serum antibodies and CRS are in line with our previous observations on the associations with serum IgA/IgG levels of *P. gingivalis* and *A. actinomycetemcomitans* and their carriage in saliva, whereas periodontitis proved to have merely a modest modifying effect [11]. Furthermore, the impact of the variation of serum antibody response between individuals in relation to genetic and immunological background, previous exposure to periodontal bacteria, and immunogenic characteristics of the antigenic challenge have been discussed [48]. IgG antibodies with a long half-life represent exposure to the bacteria; however, these antibodies are extremely stable in plasma and do not allow an estimation of the time of the exposure [13,49]. Instead, IgA antibodies with a short half-time represent a recent or continuous exposure to the indicated pathogen [50]. Therefore, it seems that saliva, in comparison to serum, would be a more suitable specimen type to recognize biomarkers of active periodontal inflammation.

The relatively large sample size (*n* = 445) and verified background data on general and oral health are the major strengths of the present study. The suitability of CRS for the evaluation of the risk of having periodontitis is established in previous studies [21,22,23]. Here we demonstrated the connection of CRS with subgingival bacterial levels as well as with antibodies against periodontal bacteria and LPS activity in saliva samples. On the other hand, a weakness of the present research is that the study population consists of patients with symptomatic heart disease, including stable coronary artery disease, acute coronary syndrome, stable or atypical chest pains, valvular heart disease, or cardiomyopathy. Thus, especially when evaluating the levels of serum antibodies, the results may not be fully applicable to a general population. Additionally, the subgingival plaque sample collected from each dental quadrant was pooled into one sample per patient. While the samples were taken from the deepest pockets, the pool may not necessarily represent the whole mouth.

## 5. Conclusions

Our new findings suggest that CRS is a reliable and suitable diagnostic tool for periodontitis. The results reveal how the salivary biomarker combination, CRS, reflects the presence of dysbiotic subgingival biofilms and the local host response to periodontal pathogens and, further, validate CRS in saliva-based diagnostics on the road to more personalized dentistry.

## Figures and Tables

**Table 1 diagnostics-10-00560-t001:** Characteristics of the study population divided into three groups on the basis of cumulative risk score (CRS) I–III.

	CRS I	CRS II	CRS III	
	Median (IQR)			*p*-Value ^1^
Age (years)	66 (10)	64 (14)	63 (13)	0.054
Number of teeth	21 (18)	24 (11)	24 (8)	**0.001**
	***n* (%)**			***p*-Value ^2^**
Sex (% of men)	70 (58.8)	119 (71.3)	114 (66.3)	0.091
Ever smoked	57 (48.3)	96 (57.5)	89 (51.7)	0.286
Diabetes	26 (21.8)	33 (20.2)	51 (30.0)	0.089
Periodontal and dental status ^3^				**<0.001**
No to mild periodontitis	88 (74.6)	125 (74.0)	102 (59.3)	
Moderate to severe periodontitis	15 (12.7)	36 (21.3)	67 (39.0)	
Edentulous	15 (12.7)	8 (4.7)	3 (1.7)	

^1^*p*-values from the Jonckheere–Terpstra test; ^2^ chi-square tests; ^3^ the classification is based on alveolar bone loss and periodontal probing pocket depth as reported in our previous study [23]. Results are presented as median values with interquartile range (IQR). Significant p-values are indicated in bold face.

**Table 2 diagnostics-10-00560-t002:** Serum antibody levels in response to five periodontal pathogens and lipopolysaccharide (LPS) levels according to the CRS.

Antigen/Measure	Antibody Class	CRS I *n* = 117	CRS II *n* = 162	CRS III n = 166	*p*-Value ^1^	OR for High CRS (95% CI), *p*-Value ^2^	
		Median (IQR)				Model 1	Model 2
***Aggregatibacter*** ***actinomycetemcomitans***	IgA	0.61 (0.61)	0.59 (0.67)	0.69 (0.69)	**0.037**	2.22 (1.19–4.18), **0.012**	2.39 (1.27–4.52), **0.007**
	IgG	0.60 (0.50)	0.69 (0.80)	0.73 (0.77)	**0.043**	1.66 (0.92–3.00), 0.094	
***Porphyromonas gingivalis***	IgA	0.24 (0.98)	0.33 (0.91)	0.86 (1.41)	**<0.001**	2.88 (1.89–4.38) **<0.001**	1.47 (0.92–2.35), 0.106
	IgG	0.84 (1.47)	0.95 (1.37)	1.71 (1.88)	**<0.001**	7.34 (3.65–14.7), **<0.001**	2.32 (1.04–5.18), **0.040**
***Porphyromonas endodontalis***	IgA	0.33 (0.55)	0.45 (0.61)	0.65 (1.26)	**<0.001**	3.52 (2.03–6.09), **<0.001**	2.95 (1.66–5.21), **<0.001**
	IgG	0.97 (1.31)	1.12 (1.17)	1.79 (1.66)	**<0.001**	7.41 (3.36–16.3), **<0.001**	6.08 (2.94–12.3), **<0.001**
***Prevotella*** ***intermedia***	IgA	0.56 (0.41)	0.56 (0.44)	0.68 (0.60)	**0.013**	3.11(1.43–6.74), **0.004**	2.59 (1.15–5.81), **0.021**
	IgG	0.44 (0.35)	0.42 (0.28)	0.46 (0.35)	0.295	2.30 (1.01–5.22), **0.047**	1.87 (0.79–4.42), 0.154
***Tannerella forsythia***	IgA	0.23 (0.16)	0.18 (0.12)	0.19 (0.14)	0.114	1.28 (0.65–2.53), 0.474	
	IgG	0.12 (0.13)	0.10 (0.11)	0.10 (0.12)	0.203	1.06 (0.57–1.97), 0.855	
**LPS**		0.46 (0.43)	0.54 (0.62)	0.53 (0.58)	0.123	1.23 (0.68–2.22), 0.497	

^1^ Jonckheere–Terpstra test; ^2^ logistic regression models, dependent variable CRS III vs. CRSI–II, odds ratio (OR) is presented/10-base logarithmically transformed unit; Model 1, adjusted for age, gender, and number of remaining teeth; Model 2, additionally adjusted for the subgingival level of the species. The antibody levels are presented as absorbance units, AU. The LPS activity is presented as endotoxin units, EU/mL. Statistically significant *p*-values are indicated by bold face.

**Table 3 diagnostics-10-00560-t003:** Saliva antibody levels in response to five periodontal pathogens and lipopolysaccharide (LPS) levels according to the CRS.

Antigen/Measure	Antibody Class	CRS I *n* = 108	CRS II *n* = 155	CRS III *n* = 157	*p*-Value ^1^	OR for High CRS (95% CI), *p*-Value ^2^	
		Median (IQR)				Model 1	Model 2
***A. actinomycetemcomitans***	IgA	2843 (2221)	3413 (3065)	4403 (5569)	**<0.001**	6.02 (3.06–11.8), **<0.001**	6.24 (3.11–12.5), **<0.001**
	IgG	4160 (4669)	7642 (15,265)	14,154 (40,410)	**<0.001**	3.28 (2.22–4.87), **<0.001**	3.26 (2.17–4.89), **<0.001**
***P. gingivalis***	IgA	3587 (5609)	5217 (5642)	9340 (12,507)	**<0.001**	6.72 (3.74–12.1), **<0.001**	4.50 (2.41–8.41), **<0.001**
	IgG	1500 (2254)	2647 (4725)	6111 (11,734)	**<0.001**	5.97 (3.72–9.61), **<0.001**	3.76 (2.26–6.25), **<0.001**
***P. endodontalis***	IgA	4809 (3249)	6971 (6315)	10,146 (9454)	**<0.001**	16.8 (7.50–37.5), **<0.001**	14.5 (6.36–32.8), **<0.001**
	IgG	1511 (1377)	2482 (2445)	4546 (5769)	**<0.001**	13.9 (7.11–27.3), **<0.001**	13.7 (6.89–27.4), **<0.001**
***P. intermedia***	IgA	2014 (1880)	3026 (2637)	3878 (4885)	**<0.001**	5.90 (3.04–11.4), **<0.001**	5.97 (3.01–11.8), **<0.001**
	IgG	1069 (637)	1534 (1755)	2547 (3284)	**<0.001**	10.2 (5.16–20.6), **<0.001**	10.0 (5.03–20.1), **<0.001**
***T. forsythia***	IgA	5893 (4254)	7849 (6614)	8359 (6605)	**<0.001**	5.43 (2.39–12.3), **<0.001**	6.92 (2.87–16.5), **<0.001**
	IgG	1448 (1069)	2208 (2712)	3956 (4397)	**<0.001**	7.83 (4.13–14.8), **<0.001**	8.81 (4.44–17.5), **<0.001**
**LPS activity**		3991 (8406)	4825 (10,816)	9065 (14,866)	**<0.001**	2.40 (1.600–3.591), **0.001**	2.42 (1.568–3.728), **<0.001** ^3^

^1^ Jonckheere–Terpstra test; ^2^ logistic regression models, dependent variable CRS III vs. CRSI–II, OR is presented/10-base logarithmically transformed unit; Model 1, adjusted for age, sex, and number of teeth; Model 2, additionally adjusted for the subgingival level of the species; ^3^ additionally adjusted for the sum of 17 Gram-negative subgingival species. The antibodies are presented as relative light units, RLU/100 ms. The LPS activity is presented as endotoxin units, EU/mL. Statistically significant *p*-values are indicated by bold face.

**Table 4 diagnostics-10-00560-t004:** Subgingival bacterial levels according to the CRS groups.

Species (Counts × 10^5^)	CRS I	CRS II	CRS III	*p*-Value ^1^	OR for High CRS (95% CI), *p*-Value ^4^
	Median (IQR)				
***A. actinomycetemcomitans***	0.32 (1.12)	0.45 (1.58)	0.79 (1.92)	**<0.001**	1.85 (1.43–2.39), **<0.001**
***P. gingivalis***	0.26 (0.77)	0.38 (1.71)	2.95 (12.7)	**<0.001**	2.70 (2.11–3.46), **<0.001**
***P. endodontalis***	0.07 (0.94)	0.22 (1.31)	1.10 (4.30)	**<0.001**	1.97 (1.57–2.47), **<0.001**
***P. intermedia***	0.47 (1.42)	0.57 (1.98)	1.55 (3.07)	**<0.001**	1.95 (1.53–2.49), **<0.001**
***T. forsythia***	1.95 (9.28)	3.47 (11.1)	12.9 (24.8)	**<0.001**	2.00 (1.63–2.46), **<0.001**
**Sum of Gram positive species ^2^**	4.24 (8.71)	4.75 (14.3)	9.82 (12.5)	**<0.001**	2.25 (1.52–3.34), **<0.001**
**Sum of Gram negative species ^3^**	25.0 (48.9)	35.0 (67.1)	70.6 (67.6)	**<0.001**	3.28 (2.10–5.13), **<0.001**

^1^ Jonckheere–Terpstra test; ^2^ sum of 11 subgingival species; ^3^ sum of 17 subgingival species; ^4^ logistic regression model, dependent variable CRS III vs. CRSI–II, OR is presented/10-base logarithmically transformed unit, adjusted for age, sex, and number of teeth. Statistically significant *p*-values are indicated by bold face.

**Table 5 diagnostics-10-00560-t005:** Association of high CRS with antibody and subgingival bacterial levels.

		Variable	Unit	OR (95% CI) for High CRS	*p*-Value ^1^
Covariates		Age	years	0.997 (0.975–1.019)	0.779
		Sex	male	0.965 (0.640–1.455)	0.865
		Number of teeth	number	1.034 (1.008–1.061)	**0.011**
		Smoking	ever	1.055 (0.703–1.583)	0.796
Model	Serum	Serum IgA burden ^2^	EU	0.910 (0.259–3.197)	0.883
		Serum IgG burden ^2^	EU	2.949 (0.536–16.23)	0.214
		Serum LPS	EU/mL	1.396 (0.681–2.860)	0.362
	Saliva	Saliva IgA burden ^2^	RLU/100 ms	7.043 (2.252–22.03)	**0.001**
		Saliva IgG burden ^2^	RLU/100 ms	3.788 (1.777–8.077)	**0.001**
		Saliva LPS	EU/mL	2.191 (1.384–3.468)	**0.001**
	Subgingival	Sum of Gram-positive species ^3^	count × 10^5^	0.641 (0.266–1.547)	0.322
		Sum of Gram-negative species ^4^	count × 10^5^	6.191 (2.098–18.27)	**0.001**

^1^ Logistic regression model, dependent variable CRS III vs. CRS I–II, model is adjusted for covariates presented above. For the bacteria, Ig, and LPS levels, OR is presented/10-base logarithmically transformed unit; ^2^ sum of antibody levels in response to *A. actinomycetemcomitans*, *P. gingivalis*, *P. endodontalis*, *P. intermedia*, and *T. forsythia*; ^3^ sum of 11 subgingival species; ^4^ sum of 17 subgingival species; ^5^ presented as counts × 10^5^ EU, ELISA units; EU/mL, endotoxin units/mL; RLU, relative light units. Statistically significant *p*-values are indicated by bold face.

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
