# Peer review of "Immunological and Microbiological Profiling of Cumulative Risk Score for Periodontitis"

_diagnostics, 2020, doi:10.3390/diagnostics10080560_

Round 1

Reviewer 1 Report

In the present study, the cumulative risk score (CRS) validity as a diagnostic marker for periodontitis is proved. To answer this question, the authors investigated the association between CRS and some microbial and inflammatory parameters. The study is well performed, and the data are clearly presented. There are a few concerns about this manuscript.

I suggest removing the last sentence in the abstract. In the present study, the authors did not specifically address periodontitis, but microbiology and immunological parameters.

More details about sample collection should be provided. Were serum and saliva samples collected after fasting time? Did participants rinse their mouths before saliva collection? Were the samples collected at similar daytime? A potential effect of these factors on the study outcome should be discussed.

The level of eluates in the saliva substantially depends on the salivary flow. The authors should either measure the salivary flow or normalize the target parameters to the total protein content. Otherwise, this point should be discussed as a limitation of the study.

In table 1, some participants were categorized as “moderate to severe periodontitis.” How was this diagnosed? Were any measurements of pocket depth, attachment loss, or bleeding index performed?

Minor comments

Line 27: use „microbial species“ instead of „microbial biomarker species.

Line 10. I assume that 28 should be written without brackets because it refers to the number of species and not to Ref. 28.

Author Response

Comment: I suggest removing the last sentence in the abstract. In the present study, the authors did not specifically address periodontitis, but microbiology and immunological parameters.

Answer: The last sentence of the abstract is now removed (Abstract).

Comment: More details about sample collection should be provided. Were serum and saliva samples collected after fasting time? Did participants rinse their mouths before saliva collection? Were the samples collected at similar daytime? A potential effect of these factors on the study outcome should be discussed.

Answer: The saliva samples were collected between 8 am and 3 pm. No fasting time or mouth rinses were applied. Keeping the collection protocol as simple as possible further promotes the utilization of salivary diagnostics for daily practice in future. As the generally accepted golden standard for saliva collection has not been launched, different collection methods cannot be seen as a limitation of the study (paragraph 2.2).

Serum samples were taken when the patient was admitted to the hospital for angiography. Fasting / non-fasting status was dependent on the indication, if the patient was suffering from an acute event (non-fasting) or if the angiography was scheduled beforehand (fasting). In addition, the samples were not taken at a specific daytime.

Comment: The level of eluates in the saliva substantially depends on the salivary flow. The authors should either measure the salivary flow or normalize the target parameters to the total protein content. Otherwise, this point should be discussed as a limitation of the study.

Answer: As specified above, simple protocols were applied for practical reasons and salivary flow was not registered (paragraph 2.2). Parameters included in the CRS (IL-1β, MMP-8, and P. gingivalis) were determined as concentrations and no normalization is needed. The saliva antibody levels are determined as relative values, but due the great dilutions used in the analyses, standardization for protein levels or salivary flow rate do not provide additional value. We have set up a method that does not require laborious collection and normalization processes and this can be seen even as a strength of the present study.

Comment: In table 1, some participants were categorized as “moderate to severe periodontitis.” How was this diagnosed? Were any measurements of pocket depth, attachment loss, or bleeding index performed?

Answer: The classification is based on alveolar bone loss and periodontal probing pocket depth, but not on bleeding index. Individuals having both alveolar bone loss from moderate to severe and at least four sites with probing pocket depth of ≥4 mm were categorized as “moderate to severe periodontitis”. In this manuscript, the periodontal status is only given to characterize the population since data about CRS vs. periodontal parameters and periodontal status has been published earlier (Salminen et al. 2014).

Minor comments

Comment: Line 27: use „microbial species” instead of „microbial biomarker species”.

Answer: The word “biomarker” is removed.

Comment: Line 101. I assume that 28 should be written without brackets because it refers to the number of species and not to Ref. 28.

Answer: The brackets are removed.

Reviewer 2 Report

Dear Authors,
The purpose of this article is to analyze the cumulative risk score (CSR) association with subgingival bacteria, salivary, and serum levels of lipopolysaccharide(LPS) and autoantibodies IgA and IgG.

The setting of the study is appropriate to previous studies (a part of a larger sample was investigated). The present study includes 455 patients.

The findings are relevant for the present research. Interestingly, serum antibody levels were independent of the subgingival bacterial level. I recommend adding a discussion connected to this (line 242).

The association observed between high CRS with antibodies against P. gingivalis and P. endodontalis brings interesting data and is a good future line for research.
The conclusion is brief and summarizes the main points of the research.

The article is concise and well-written and needs minor revision.

Author Response

Comment: The findings are relevant for the present research. Interestingly, serum antibody levels were independent of the subgingival bacterial level. I recommend adding a discussion connected to this (line 242).

Answer: The sentence on line 242 relates to the association analysis. This topic is discussed later in the same chapter: “Notably, salivary IgA/IgG burden against these microbial biomarker species presented an independent and strong association with CRS even in models where the bacterial subgingival levels were used as covariates. This indicates immune hyperresponsiveness to dysbiosis typical for the pathogenesis of periodontitis [46].” Furthermore, the Discussion continues: “Also, the impact of the variation of serum antibody response between individuals in relation to genetic and immunological background, previous exposure to periodontal bacteria, and immunogenic characteristics of the antigenic challenge have been discussed [48]”. This is probably explained also by the fact that subgingival plaque sampling is not a quantitative method since only one sample per quadrant is collected and pooled.

Comment: The article is concise and well-written and needs minor revision.

Answer: The minor styling and grammatical revision was performed throughout the manuscript by professor Persson and can be seen via ‘track changes’ function.